

# Prognostic significance of PI3K/AKT/mTOR signaling pathway members in clear cell renal cell carcinoma

Demin Fan[1,*], Qiang Liu[2,*], Fei Wu[1], Na Liu[3], Hongyi Qu[4], Yijiao Yuan[4], Yong Li[5], Huayu Gao[4], Juntao Ge[4], Yue Xu[4], Hao Wang[4], Qingyong Liu[1] and Zuohui Zhao[4]

[1] Department of Urology, The First Affiliated Hospital of Shandong First Medical University, Jinan, China
[2] Laboratory of Microvascular Medicine, The First Affiliated Hospital of Shandong First Medical University, Jinan, China
[3] Department of Hematology, Qilu Hospital of Shandong University, Jinan, China
[4] Department of Pediatric Surgery, The First Affiliated Hospital of Shandong First Medical University, Jinan, China
[5] Department of Urology, Shandong Yuncheng County Chinese Medicine Hospital, Heze, China
[*] These authors contributed equally to this work.

Corresponding author
Zuohui Zhao,
zhaozuohui@sdhospital.com.cn,
zhaozuohui@126.com

## ABSTRACT

**Background**. Renal cell carcinoma (RCC) is a fatal disease, in which the PI3K/AKT/mTOR signaling pathway serves an important role in the tumorigenesis. Previous studies have reported the prognostic significance of PI3K/AKT/mTOR signaling pathway members in RCC; however, there is insufficient evidence to date to confirm this. Thus, the present study aimed to systematically investigate the prognostic roles of multiple PI3K/AKT/mTOR signaling proteins in clear cell RCC (ccRCC) using online large-scale databases.

**Methods**. The mRNA expression profiles of PI3K/AKT/mTOR signaling pathway proteins *PTEN, PIK3CA, PIK3CB, PIK3CD, PIK3CG, AKT1, AKT2, AKT3* and *mTOR* were investigated using the Gene Expression Profiling Interactive Analysis (GEPIA) and Oncomine databases, and the protein expression levels of PI3K, AKT and mTOR were detected using western blotting (WB) analysis. In addition, the correlation between mRNA or protein expression levels and the prognostic significance was analyzed using the Kaplan-Meier (K-M) plotter ($n = 530$), the Human Protein Atlas (HPA; $n = 528$) and The Cancer Protein Atlas (TCPA; $n = 445$) databases.

**Results**. The GEPIA revealed that the mRNA expression of major PI3K/AKT/mTOR pathway members, including *PTEN, PIK3CA, PIK3CB, AKT1, AKT2* and *AKT3,* were negatively correlated with ccRCC stages ($P < 0.05$), though most of their mRNA and protein expression levels were notsignificantly different between ccRCC and normal tissues using GEPIA, Oncomine and WB analyses ($P < 0.05$). Meanwhile, using the K-M plotter and HPA prognostic analysis, it was found that the mRNA expression levels of the majority of the PI3K/AKT/mTOR signaling pathway members, including *PTEN, PIK3CA, PIK3CB, PIK3CG, AKT3* and *mTOR* were positively correlated with overall survival (OS), whereas *PIK3CD* mRNA expression was negatively correlated with OS ($P < 0.05$). Furthermore, TCPA prognostic analysis observed that several of the key molecules of the PI3K/AKT/mTOR signaling pathway [*PTEN, p-AKT (S473)* and *p-mTOR (S2448)*] were also positively correlated with OS in patients with ccRCC ($P < 0.05$). In conclusion, the present study suggested that several members of the

PI3K/AKT/mTOR signaling pathway, especially *PTEN*, may be favorable prognostic factors in ccRCC, which indicated that the PI3K/AKT/mTOR signaling pathway may be implicated in ccRCC initiation and progression.

## INTRODUCTION

Kidney cancer is a high-risk cancer that demonstrates a high mortality rate, accounting for an estimated 73,750 newly diagnosed cases and 14,830 cancer-related deaths in the United States in 2020 (*Siegel, Miller & Jemal, 2020*). Clear cell renal cell carcinoma (ccRCC), the most common subtype of kidney cancer, accounts for the majority of kidney cancer-related deaths (*Rini, Campbell & Escudier, 2009*). Localized RCC is commonly treated using nephrectomy and exhibits good prognosis; however, due to the resistance to traditional chemotherapy and radiotherapy, advanced RCC is considered a fatal malignancy with a poor overall survival (OS) (*Rini, Campbell & Escudier, 2009*). With the advances in tumor molecular genetics, the treatment for advanced RCC, in particular for metastatic RCC (mRCC), has evolved from traditional cytotoxic therapy to molecular targeted therapy, for example, sunitinib and everolimus, which have significantly prolonged the survival time for patients with RCC (*Han et al., 2017*). However, the molecular targeted therapy for RCC remains limited due to the lack of molecular targets. Thus, there is an urgent requirement to further investigate the molecular mechanisms that drive RCC initiation and progression, which will help identify potential therapeutic targets and prognostic markers for RCC therapy.

Apart from the well-known von Hippel-Lindau (VHL) gene, genetic mutations in members of the PI3K/AKT/mTOR signaling pathway are also frequently observed in RCC, which promotes the hyperactivation of the PI3K/AKT/mTOR signaling cascade (*Han et al., 2017*). The PI3K/AKT/mTOR signaling pathway, which is inhibited by phosphatase and tensin homolog (PTEN), is transduced through three checkpoint proteins or protein complexes: PI3K, AKT and mTOR. PTEN, PI3K, AKT and mTOR participate in the regulation of multiple biological functions, including cell survival, metabolism and tumorigenesis under context-specific physiological and pathological conditions (*Cargnello, Tcherkezian & Roux, 2015*; *Guo et al., 2015*). PTEN is a tumor suppressor gene, which is frequently mutated in tumors (*Hager et al., 2011*). PI3K consists of four isoforms: PIK3CA, PIK3CB, PIK3CD and PIK3CG. The activation of PI3K phosphorylates phosphatidylinositol-4,5-bisphosphate (PIP2) to generate phosphatidylinositol-3,4,5-trisphosphate (PIP3), which subsequently recruits AKT to the plasma membrane, where AKT is activated by 3-phosphoinositide-dependent protein kinase (PDK) phosphorylation on Thr308. This process can be reversed by PTEN, a phosphatase that can transform PIP3 to inactivated PIP2. mTOR is a 289 kDa serine/threonine kinase that encompasses two functionally distinct protein complexes: mTOR complex 1 (mTORC1) and mTORC2

(*Guo et al., 2015*; *Zhang et al., 2017*). Activated AKT phosphorylates mTORC1 on Ser2,448 to activate its kinase activity, which then leads to the activation of multiple anabolic biosynthetic pathways that control cell proliferation (*Zhang et al., 2017*). PI3K/AKT/mTOR signaling pathway dysregulation is frequently identified in patients with RCC (*Han et al., 2017*), and inhibitors of mTOR, including everolimus and temsirolimus, have proven efficacious in mRCC (*Escudier et al., 2014*; *Ghidini et al., 2017*). In fact, previous studies have reported the prognostic value of PI3K/AKT/mTOR signaling pathway-related proteins; however, the majority of these studies were based on small-scale specimens and the conclusions lacked consistency (*Hager et al., 2009*; *Kim et al., 2017*; *Kruck et al., 2010*; *Liontos et al., 2017*; *Merseburger et al., 2008*; *Rausch et al., 2019*; *Zhu et al., 2015*). Thus, it is necessary to investigate the prognostic role of the PI3K/AKT/mTOR signaling pathway using large-scale databases.

By taking advantage of next-generation sequencing and multi-omics, a large number of potential biomarkers and therapeutic targets have emerged in recent years. The Cancer Genome Atlas (TCGA) (*Han et al., 2017*), The Cancer Proteome Atlas (TCPA) (*Li et al., 2013*), as well as several other large-scale cancer genome tools such as the Gene Expression Profiling Interactive Analysis (GEPIA) (*Tang et al., 2019*), the Kaplan–Meier (K-M) plotter (*Sui et al., 2019*) and the Human Protein Atlas (HPA) (*Zhu et al., 2018*), have been comprehensively used to investigate the molecular profiles of several types of cancer. Nevertheless, only a few potentially useful prognostic biomarkers have been reported for RCC to date; for example, through screening the TCGA database, *Han et al. (2017)* discovered that numerous protein biomarkers, such as *fatty acid synthase (FASN)* and *AKT3* were prognostic factors for the survival of patients with ccRCC; and using the Oncomine and K-M plotter databases, our previous study reported that several mRNA biomarkers, such as *FASN* and *ATP citrate lyase (ACLY)*, may be prognostic factors for ccRCC (*Zhao et al., 2019*). In the present study, the mRNA and some protein expression profiles of PI3K/AKT/mTOR signaling pathway members were compared between ccRCC and normal kidney tissues using the GEPIA, Oncomine and western blotting (WB) analyses, and their prognostic significance in ccRCC was subsequently systematically analyzed using the K-M plotter, HPA and TCPA databases (Fig. 1). In addition, the correlation between these PI3K/AKT/mTOR signaling pathway members and specific clinicopathological parameters of cancer, such as cancer stage, was also explored. In conclusion, the present study provided a comprehensive overview of the OS biomarkers found in the PI3K/AKT/mTOR signaling pathway, which highlighted the potential role of this pathway in ccRCC progression and targeted therapy.

## MATERIALS & METHODS

### GEPIA analysis

The GEPIA server (http://gepia2.cancer-pku.cn/), which was based on the UCSC Xena project, was used to analyze the mRNA expression status of the following nine members of the PI3K/AKT/mTOR signaling cascade in ccRCC, as described previously (*Tang et al., 2019*): *PTEN, PI3K* (including *PIK3CA, PIK3CB, PIK3CD* and *PIK3CG*), *AKT* (including *AKT1, AKT2* and *AKT3*) and *mTOR*. Briefly, the mRNA expression data of 523 ccRCC
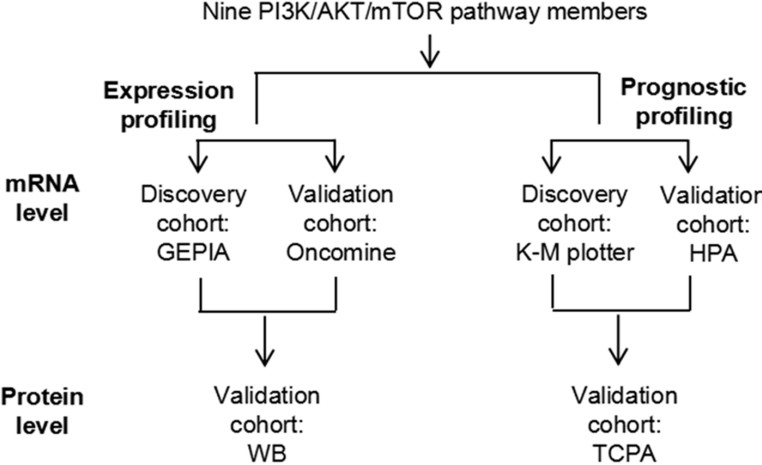

**Figure 1** A schematic diagram to investigate the expression and prognostic profiling of PI3K/AKT/mTOR signaling pathway members in ccRCC.

and 100 normal kidney specimens from TCGA and Genotype-Tissue Expression (GTEx) were loaded into the server to analyze the differentially expressed genes between the two groups. The RNA-seq results were reported as the number of transcripts per million (TPM). The following filter indexes were set: expression mode was expression DIY, dataset was ccRCC, log-scale was log2 (TPM + 1), and *P*-value cut-off was 0.05. Then the mRNA expression parameters, including sample size and statistical box plot were displayed. For consistency analysis, the mRNA expression levels of the nine genes in 523 ccRCC and 72 matched normal specimens from TCGA, and their correlationship between the genes and the pathological stages in 523 ccRCC specimens were also analyzed.

## Oncomine analysis

The Oncomine database (https://www.oncomine.org) was used to validate the mRNA expression levels of the PI3K/AKT/mTOR pathway members, as previously described (*Zhao et al., 2018*). After indexes were set: differential analysis (cancer versus normal tissue), cancer type (kidney cancer), and data type (mRNA), their differential expression parameters of the individual genes, such as sample size, *P*-value, fold changes and statistical box plot, were dispalyed.

## WB analysis

All the patients signed the written informed consents before enrollment, and the procedures were approved by the ethical committees of The First Affiliated Hospital of Shandong First Medical University (No. 2017-S007). And 3 cases of ccRCC tissues (including cancer and adjacent non-cancerous), which were collected from 3 patients (2 males and 1 females, age from 54 to 65, Fuhrman nuclear grading with 1 G2 + 2 G3, TNM staging with 2 T1 + 1 T2) (*Edge & Compton, 2010*), were used for WB analysis as described previously (*Zhao et al., 2017*). After incubated with the corresponding primary antibodies: anti-PI3K (1:1,000, rabbit, #4249; Cell signaling Technology, Danvers, MA), anti-AKT (1:1,000, rabbit, #

4691; Cell signaling Technology, Danvers, MA), anti-mTOR (1:2,000, rabbit, ab2732; Abcam, Cambridge, MA), and anti-β-Actin (1:1,000, mouse, BM0627; BOSTER, Beijing, China), horseradish peroxidase (HRP) conjugated secondary antibodies (1:5,000, rabbit, SA00001-2 & mouse, SA00001-1; Proteintech, Wuhan, China) were used to detect the proteins. Then the blots were visualized by enhanced chemiluminescence (Amersham Imager 600, GE Healthcare, Marlborough, MA). Protein ladder (MAN0011772, Thermo Scintific, Waltham, MA) was used to label the corresponding proteins, and Image J software was used for quantification.

## K-M plotter mRNA survival analysis

The K-M plotter (http://www.kmplot.com), which contained the RNA-seq data from Gene Expression Omnibus (GEO), Cancer Biomedical Informatics Grid (caBIG) and TCGA, was used to analyze the correlation between the mRNA expression levels of each PI3K/AKT/mTOR signaling pathway member and the prognosis of patients with ccRCC ($n = 530$), as described previously (*Sui et al., 2019*). First, the filter indexes were set: the pathological type was ccRCC, and auto select best cut-off as the default cut-off. Then the nine PI3K/AKT/mTOR signaling pathway genes (*PTEN, PIK3CA, PIK3CB, PIK3CD, PIK3CG, AKT1, AKT2, AKT3* and *mTOR*) were loaded into the online server, and the K-M survival plots were used to identify their correlation with the OS for discovery analysis. The hazard ratio (HR), with 95% confidence intervals (CI), and the log-rank $P$-value were calculated automatically on the webpage after ≤150 months follow-up. The relationship between the gene expression levels and clinicopathological parameters, for example, sex, pathological stages and grades, of ccRCC, were also investigated.

## HPA mRNA survival analysis

The HPA database (http://www.proteinatlas.org), which retrieved the RNA-seq data from the Genomic Data Commons (GDC) Data Portal, was used to validate the prognosis of PI3K/AKT/mTOR signaling pathway genes in patients with ccRCC (*Zhu et al., 2018*). In the public database, the filter indexes were set: the pathological type was ccRCC, best expression cut-off as the default cut-off, then all the nine aforementioned genes were loaded into the database and the OS from the TCGA was used for validation analysis ($n = 528$). The K-M plot and log-rank $P$-value were also calculated accordingly after ≤150 months follow-up.

## TCPA protein survival analysis

TCPA database (https://tcpaportal.org/tcpa/) was used to analyze the correlation between the PI3K/AKT/mTOR signaling pathway protein expression levels and the prognosis of patients with ccRCC (*Li et al., 2013*). Briefly, seven proteins, including PTEN, PIK3CA, AKT, mTOR and three proteins modified by phosphorylation, phosphorylated (p)-AKT (S473), p-AKT (T308) and p-mTOR (S2448), were evaluated using TCPA database ($n = 445$) after ≤150 months follow-up. The log-rank and univariate Cox $P$-values were calculated automatically. The relationship between the protein expression levels and the clinicopathological parameters, for instance, the pathological stage and Fuhrman grade of ccRCC, were also investigated.

## Statistical analysis

Statistical analysis was performed using SPSS 21.0 software (IBM Corp.). For GEPIA analysis, the mRNA expression levels of PI3K/AKT/mTOR signaling pathway members in ccRCC and normal tissues were analyzed using one-way ANOVA. For Oncomine and WB analyses, the PI3K/AKT/mTOR expression levels were analyzed using a two-tailed Student's $t$-test. For the K-M plotter and HPA analyses, the correlation between the mRNA expression levels and OS was calculated using K-M curves and the log-rank test. For TCPA analysis, the correlation between protein expression levels and OS was calculated using the log-rank test and a univariate Cox Proportional Hazards regression model. $P < 0.05$ were considered to indicate a statistically significant difference.

# RESULTS

## GEPIA, Oncomine and WB analyses of the expression levels of PI3K/AKT/mTOR signaling pathway members between ccRCC and normal tissues

First, the mRNA expression profiles of the PI3K/AKT/mTOR signaling pathway members were analyzed using GEPIA as described previously (*Tang et al., 2019*). The RNA-Seq data from TCGA and GTEx were used to compare the expression levels of nine PI3K/AKT/mTOR signaling pathway members (*PTEN, PIK3CA, PIK3CB, PIK3CD, PIK3CG, AKT1, AKT2, AKT3* and *mTOR*) between ccRCC and normal kidney tissues. The mean mRNA expression levels of these genes were not significantly different between the 523 ccRCC and 100 normal tissues ($P > 0.05$; Fig. 2); however, both *PIK3CD* and *PIK3CG* displayed an upward trend while *mTOR* displayed a decreasing trend, but not significant expression levels in patients with ccRCC ($P > 0.05$). Similar results were obtained between 523 ccRCC and 72 matched normal samples using GEPIA analysis (Table S1).

Regarding the association of mRNA expression with pathological stages of the ccRCC patients, the detailed expression levels of the PI3K/AKT/mTOR members were further investigated in the 523 patients using GEPIA. The violin plots showed their expression in each stage, which illustrated the mRNA expression levels of six genes, i.e., *PTEN* ($P = 0.002$), *PIK3CA* ($P < 0.001$), *PIK3CB* ($P = 0.010$), *AKT1* ($P < 0.001$), *AKT2* ($P < 0.001$) and *AKT3* ($P < 0.001$), were negatively correlated with their stages (Fig. 3).

Then Oncomine database (U133A/B microarray, 10 ccRCC and 10 matched normal kidney specimens) was used to validate their differential expression (Table S2), which demonstrated that most of the PI3K/AKT/mTOR members were not significantly changed between ccRCC and normal specimens ($P > 0.05$) except PTEN (1.77-fold increase in ccRCC, $P < 0.001$), PIK3CA (1.56-fold increase, $P = 0.010$) and AKT1 (1.27-fold increase, $P = 0.005$).

Finally, further WB analysis displayed that the protein expression levels of PI3K, AKT and mTOR were not significantly changed between ccRCC and normal specimens ($P = 0.774$, $P = 0.585$, $P = 0.480$, respectively; Fig. 4), which also consistent with the mRNA expression of PI3K/AKT/mTOR signaling members.

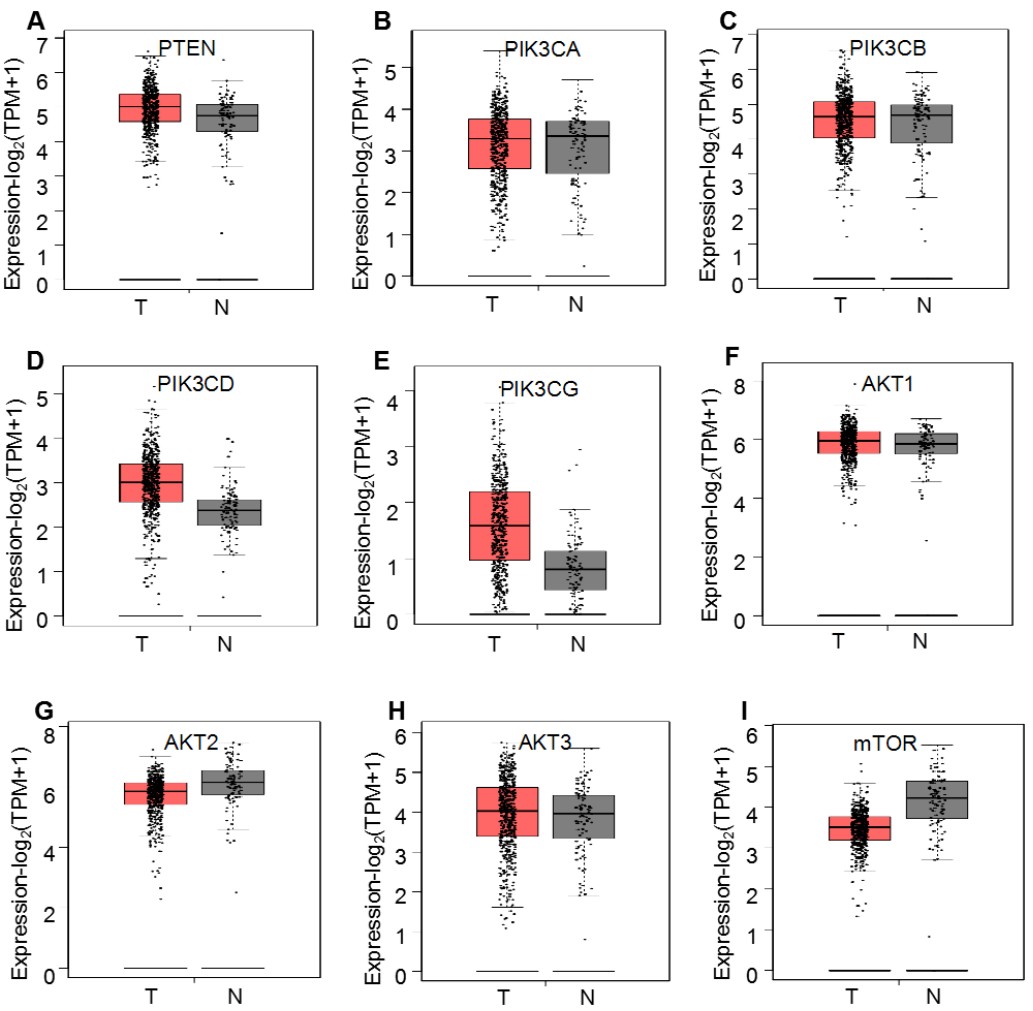

**Figure 2** **GEPIA mRNA expression profiles of PI3K/AKT/mTOR signaling pathway proteins in 523 ccRCC (T) and 100 normal ones (N), which was downloaded from TCGA and GTEx databases.** Box plots demonstrate the relative mRNA expression levels (log2 (TPM+1)) of (A) *PTEN*, (B) *PIK3CA*, (C) *PIK3CB*, (D) *PIK3CD*, (E) *PIK3CG*, (F) *AKT1*, (G) *AKT2*, (H) *AKT3* and (I) *mTOR*, which were not significantly altered between the tissues ($P > 0.05$).

## K-M plotter and HPA prognostic analysis of the PI3K/AKT/mTOR signaling pathway members in ccRCC

For survival analysis, the public online resource K-M plotter was used to calculate the clinical significance of individual genes compared with OS in 530 patients with ccRCC (Fig. 5; Table 1). The results demonstrated that increased mRNA expression levels of *PTEN* [HR 95% CI = 0.56 (0.41–0.75); $P < 0.001$] and *mTOR* [HR 95% CI = 0.54 (0.38–0.78); $P < 0.001$] were associated with improved OS in patients with ccRCC. For the multiple *AKT* genes, *AKT3* [HR 95% CI = 0.44 (0.32–0.62); $P < 0.001$] was observed to be a favorable factor; however *AKT1* [HR 95% CI = 0.76 (0.56–1.02); $P = 0.068$] and *AKT2* [HR 95% CI = 1.34 (0.99–1.82); $P = 0.055$] were not. As for the *PI3K* genes, the majority were identified as protective factors, including *PIK3CA* [HR 95% CI = 0.52 (0.38–0.70);

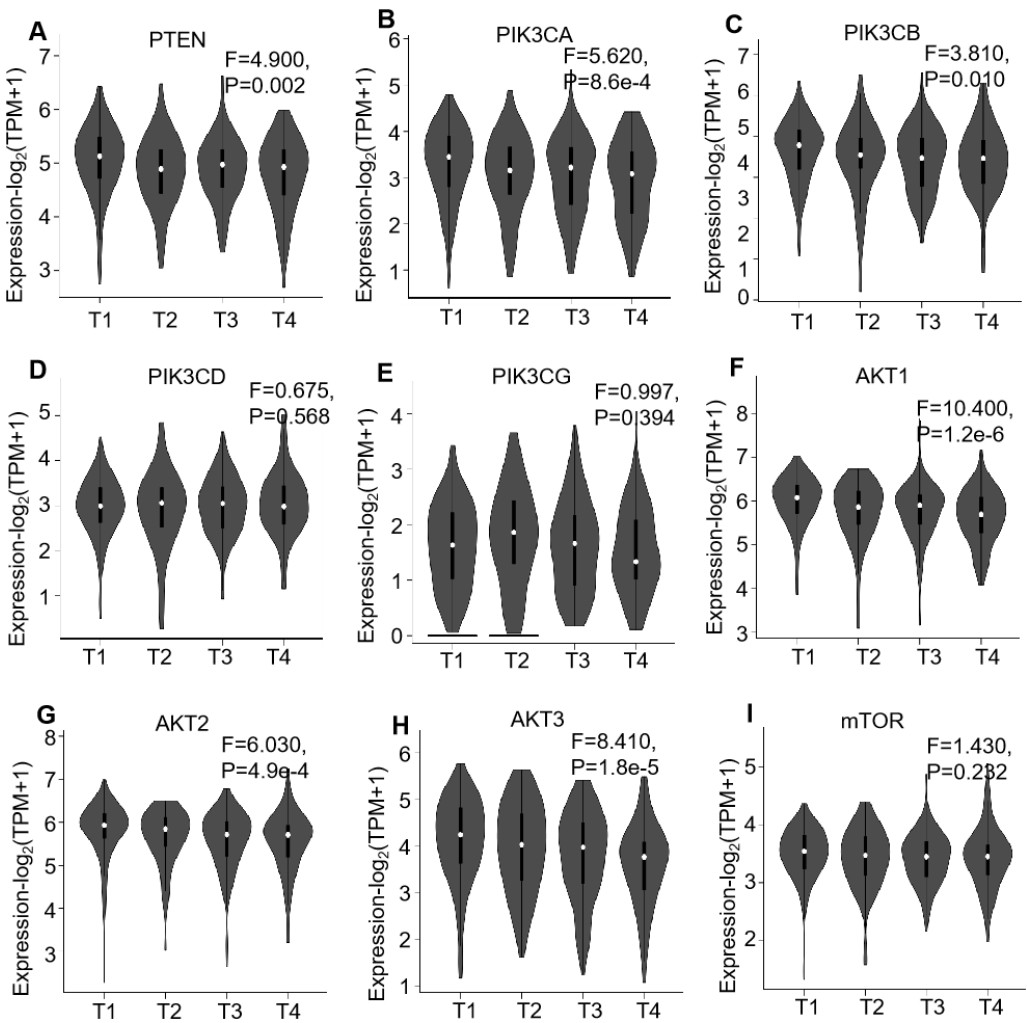

**Figure 3** **GEPIA mRNA expression profiles of PI3K/AKT/mTOR pathway members in 523 ccRCC tissues, which investigated the association of their mRNA expression with pathological stages of ccRCC patients.** Violin plots demonstrate the relationship between mRNA expression levels (log2 (TPM+1)) of (A) *PTEN*, (B) *PIK3CA*, (C) *PIK3CB*, (D) *PIK3CD*, (E) *PIK3CG*, (F) *AKT1*, (G) *AKT2*, (H) *AKT3* and (I) *mTOR*, which were not significantly altered between the tissues ($P > 0.05$).

$P < 0.001$], *PIK3CB* [HR 95% CI = 0.50 (0.37–0.68); $P < 0.001$] and *PIK3CG* [HR 95% CI = 0.64 (0.47–0.87); $P = 0.004$], whereas *PIK3CD* demonstrated the opposite effect [HR 95% CI = 1.38 (1.01–1.91); $P = 0.046$]. These results suggested that different members of AKT or PI3K may serve distinct roles in the prognosis of ccRCC.

In addition to investigating OS, cancer staging, sex and grading were further investigated to determine the correlation between mRNA expression levels and OS in ccRCC patients. According to the TNM staging system (*Edge & Compton, 2010*), the ccRCC samples were grouped into four stages and the correlation between mRNA expression levels and OS in different pathological stages of ccRCC patients was analyzed (Table 2). It was observed that *PTEN*, *PIK3CA* and *PIK3CB* were positively correlated with a favorable OS in stage I,

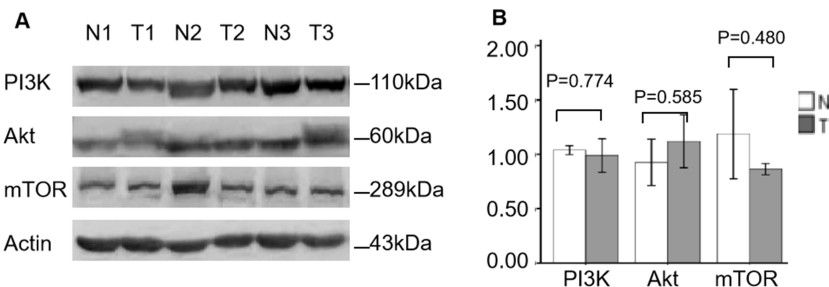

**Figure 4** **Validation of PI3K, AKT and mTOR protein expression in ccRCC by WB.** (A) WB for PI3K, AKT and mTOR expression of three individual ccRCC (T) and adjacent normal tissues (N). Actin as a loading control. (B) The statistic results of gray value ratios of the WB bands in ccRCC and normal tissues.

III and IV patients with ccRCC [HR 95% CI = 0.46 (0.25–0.85) (stage I); 0.44 (0.25–0.79) (stage III); 0.51 (0.30–0.87) (stage IV) for *PTEN*; 0.48 (0.26–0.88) (stage I); 0.51 (0.29–0.89) (stage III); 0.61 (0.37–1.00) (stage IV) for *PIK3CA*; and 0.43 (0.23–0.81) (stage I); 0.40 (0.22–0.70) (stage III); 0.41 (0.25–0.67) (stage IV) for *PIK3CB*, respectively]. Furthermore, increased *PIK3CG* expression level was correlated with an improved OS in stage I, II and III patients with ccRCC [HR 95% CI = 0.53 (0.28–1.00); 0.29 (0.08–1.00); 0.31 (0.17–0.54), respectively]. Similar results were achieved for *AKT3 and mTOR; AKT3* was found to be positively correlated with a favorable prognosis in stage III and IV patients with ccRCC [HR 95% CI = 0.45 (0.25–0.81); 0.50 (0.27–0.94), respectively] and the overexpression of *mTOR* was correlated with an improved OS in stage I and II patients with ccRCC [HR 95% CI = 0.30 (0.12–0.77); 0.24 (0.06–0.86), respectively]. *AKT1* was discovered to be a favorable factor in stage I [HR 95% CI = 0.44 (0.24–0.80)], whereas it found to be a poor factor in stage II and III patients with ccRCC [HR 95% CI = 3.00 (0.98–9.20); 2.15 (1.14–4.07), respectively]. *PIK3CD* mRNA expression levels were associated with a favorable OS in stage III [HR 95% CI = 0.55 (0.31–0.96)], but a worse OS in stage IV [HR 95% CI = 2.42 (1.25–4.68)] patients with ccRCC. *AKT2* mRNA expression was not correlated with the tumor stages of patients with ccRCC ($P > 0.05$). These results suggested that the prognostic role of PI3K/AKT/mTOR signaling pathway members may be markedly influenced by TNM stages.

Regarding the association of these genes with the sex of patients, the prognostic values of individual genes were further investigated in patients with ccRCC. As illustrated in Table 3, *PTEN, PIK3CA, PIK3CB, PIK3CG* and *AKT3* were positively correlated with a favorable OS in both females [HR 95% CI = 0.35 (0.21–0.59) for *PTEN*; 0.27 (0.15–0.51) for *PIK3CA*; 0.45 (0.27–0.76) for *PIK3CB*; 0.49 (0.30–0.82) for *PIK3CG*; and 0.32 (0.20–0.54) for *AKT3*, respectively] and males [HR 95% CI = 0.53 (0.36–0.77) for *PTEN*; 0.57 (0.39–0.82) for *PIK3CA*; 0.48 (0.33–0.70) for *PIK3CB*; 0.68 (0.47–0.99) for *PIK3CG*; and 0.48 (0.31–0.72) for *AKT3*, respectively]. *mTOR* was positively correlated with an improved OS in female patients with ccRCC [HR 95% CI = 0.27 (0.14–0.52)], whereas *AKT2* was positively correlated with poor OS in males [HR 95% CI = 1.71 (1.15–2.55)]. Moreover, *PIK3CD* was positively correlated with a favorable OS in females [HR 95% CI = 0.57 (0.34–0.97)],

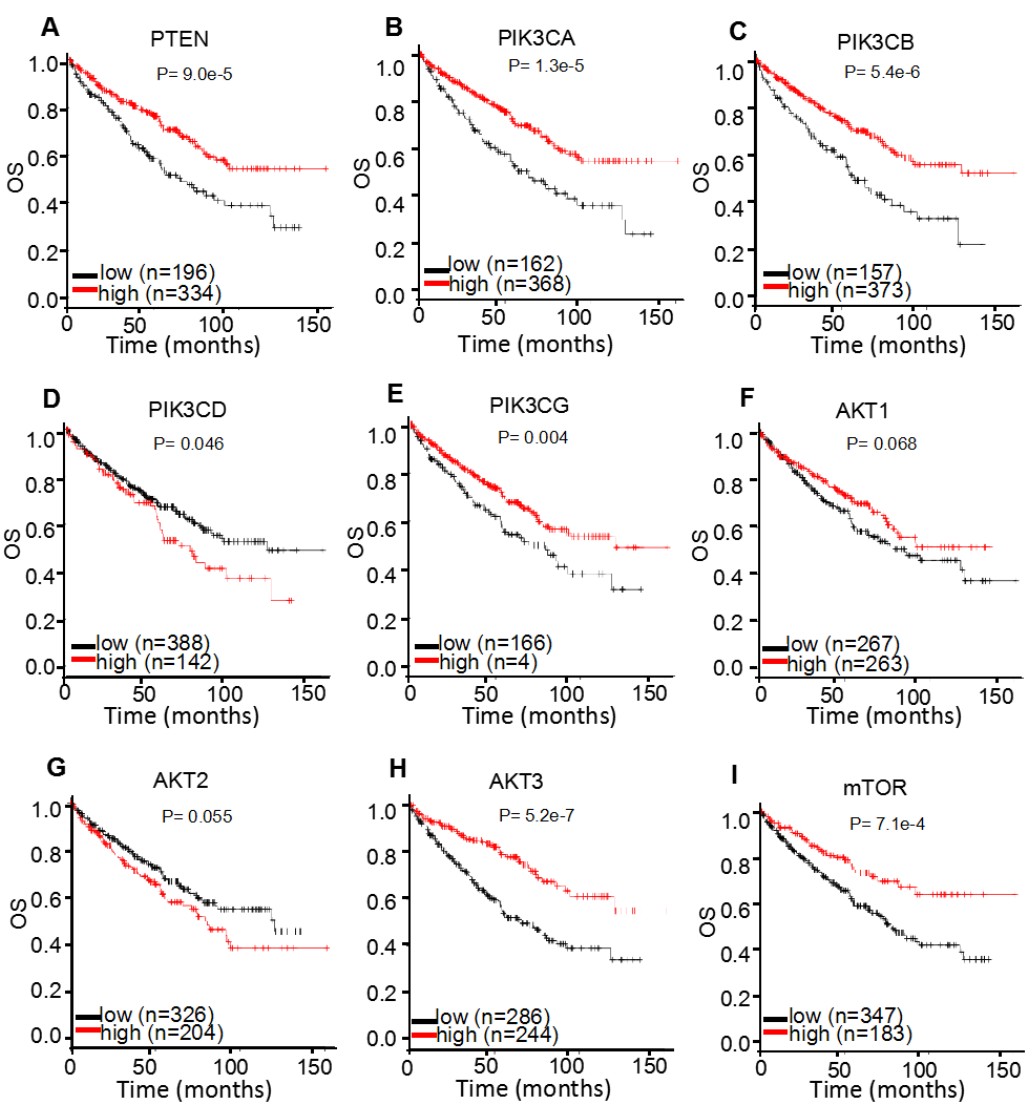

**Figure 5** K-M plotter mRNA analysis displaying the overall survival curves of PI3K/AKT/mTOR signaling pathway members. The overall survival curves of (A) *PTEN*, (B) *PIK3CA*, (C) *PIK3CB*, (D) *PIK3CD*, (E) *PIK3CG*, (F) *AKT1*, (G) *AKT2*, (H) *AKT3* and (I) *mTOR* expression levels in 530 patients with clear cell renal cell carcinoma.

but poor OS in male patients with ccRCC [HR 95% CI = 1.81 (1.22–2.68)]. These results suggested that sex may affect the prognostic role of PI3K/AKT/mTOR signaling pathway members in patients with ccRCC.

Similarly, the prognostic value of the genes in different grades of ccRCC was also evaluated. As illustrated in Table S3 , the OS of the nine genes in different grades were consistent with the total OS.

For HPA validation analysis, the prognostic significance of the nine genes was examined, which found that the majority of genes demonstrated identical prognosis (OS) following the same follow-up period in another 528 patients with ccRCC (Fig. 6). Specifically, increased

**Table 1** Correlation of the PI3K/AKT/mTOR mRNA expression with OS in ccRCC patients (K-M plotter, $n = 530$).

| Protein | Gene | HR (95% CI) | P- value |
|---------|------|-------------|----------|
| PTEN | PTEN | 0.56 (0.41–0.75) | <0.001 |
| PI3K | PIK3CA | 0.52 (0.38–0.70) | <0.001 |
| | PIK3CB | 0.50 (0.37–0.68) | <0.001 |
| | PIK3CD | 1.38 (1.01–1.91) | 0.046 |
| | PIK3CG | 0.64 (0.47–0.87) | 0.004 |
| AKT | AKT1 | 0.76 (0.56–1.02) | 0.068 |
| | AKT2 | 1.34 (0.99–1.82) | 0.055 |
| | AKT3 | 0.44 (0.32–0.62) | <0.001 |
| mTOR | mTOR | 0.54 (0.38–0.78) | <0.001 |

expression levels of *PTEN* ($P < 0.001$), *PIK3CA* ($P < 0.001$), *PIK3CB* ($P < 0.001$), *PIK3CG* ($P = 0.004$), *AKT3* ($P < 0.001$) and *mTOR* ($P = 0.001$) were correlated with a significantly improved OS in 528 patients with ccRCC, whereas the overexpression of *PIK3CD* mRNA ($P = 0.048$) was significantly correlated with poor OS. *AKT1* mRNA expression levels demonstrated no significant correlation with the prognosis of patients with ccRCC ($P = 0.082$), with the K-M plotter results providing similar findings. In addition, HPA analysis identified *AKT2* as a hazard factor ($P = 0.004$), but it was not correlated with patients' prognosis ($P = 0.055$), which was assessed using K-M plotter analysis.

## TCPA prognostic analysis of the protein expression levels of PI3K/AKT/mTOR signaling pathway members in ccRCC

Since the activity of the PI3K/AKT/mTOR signaling cascade is largely regulated through phosphorylation events, the prognostic roles of core executors of the PI3K/AKT/mTOR signaling pathway, with or without phosphorylation, were investigated in patients with ccRCC. For this purpose, 445 cases of ccRCC from TCPA database were used for the statistical analysis (Fig. 7; Table 4). It was discovered that the higher expression levels of *PTEN* (log-rank $P < 0.001$; Cox $P < 0.001$), p-AKT (S473; log-rank $P = 0.047$; Cox $P = 0.002$) and p-mTOR (S2448; log-rank $P = 0.004$; Cox $P < 0.001$) were associated with an improved OS in patients with ccRCC, which is consistent with previous findings (*Zhang et al., 2017*). Although *AKT* and *mTOR* were not recognized as significant prognostic factors by K-M plot analysis (log-rank $P = 0.263$ for AKT; $P = 0.543$ for mTOR), they were identified as favorable prognostic factors by univariate Cox Proportional Hazards model analysis (Cox $P = 0.003$ for AKT; $P = 0.041$ for mTOR). However, PIK3CA and p-AKT (T308) expression levels were not correlated with tumor prognosis in patients with ccRCC ($P > 0.05$).

As tumor staging and grading could markedly influence the statistical results of prognosis, correlation analysis between protein expression levels and ccRCC stages or grades were further performed (Table 4). The expression levels of *PTEN, AKT* and *p-AKT (S473)* were positively correlated with both tumor stages ($P < 0.001$; $P = 0.045$ and $P < 0.001$, respectively) and tumor grades ($P < 0.001$; $P < 0.001$ and $P < 0.001$, respectively) in patients with ccRCC, which is consistent with previous studies

**Table 2  Correlation of the PI3K/AKT/mTOR mRNA expression with OS in different clinical stage of ccRCC patients (K-M plotter, $n = 530$).**

| Gene | Stage | n[a] | HR (95% CI) | P- value |
|---|---|---|---|---|
| PTEN | I | 265 | 0.46 (0.25–0.85) | 0.011 |
| | II | 57 | 3.28 (1.04–10.37) | 0.034 |
| | III | 123 | 0.44 (0.25–0.79) | 0.005 |
| | IV | 82 | 0.51 (0.30–0.87) | 0.012 |
| PIK3CA | I | 265 | 0.48 (0.26–0.88) | 0.016 |
| | II | 57 | 2.08 (0.67–6.48) | 0.200 |
| | III | 123 | 0.51 (0.29–0.89) | 0.017 |
| | IV | 82 | 0.61 (0.37–1.00) | 0.049 |
| PIK3CB | I | 265 | 0.43 (0.23–0.81) | 0.007 |
| | II | 57 | 2.14 (0.47–9.73) | 0.310 |
| | III | 123 | 0.40 (0.22–0.70) | 0.001 |
| | IV | 82 | 0.41 (0.25–0.67) | <0.001 |
| PIK3CD | I | 265 | 1.41 (0.74–2.70) | 0.300 |
| | II | 57 | 2.25 (0.69–7.33) | 0.170 |
| | III | 123 | 0.55 (0.31–0.96) | 0.033 |
| | IV | 82 | 2.42 (1.25–4.68) | 0.007 |
| PIK3CG | I | 265 | 0.53 (0.28–1.00) | 0.045 |
| | II | 57 | 0.29 (0.08–1.00) | 0.046 |
| | III | 123 | 0.31 (0.17–0.54) | <0.001 |
| | IV | 82 | 0.59 (0.34–1.03) | 0.062 |
| AKT1 | I | 265 | 0.44 (0.24–0.80) | 0.006 |
| | II | 57 | 3.00 (0.98–9.20) | 0.044 |
| | III | 123 | 2.15 (1.14–4.07) | 0.015 |
| | IV | 82 | 1.28 (0.78–2.10) | 0.330 |
| AKT2 | I | 265 | 1.65 (0.91–2.99) | 0.093 |
| | II | 57 | 2.22 (0.76–7.14) | 0.130 |
| | III | 123 | 1.58 (0.90–2.78) | 0.110 |
| | IV | 82 | 1.61 (0.96–2.68) | 0.067 |
| AKT3 | I | 265 | 0.56 (0.31–1.02) | 0.054 |
| | II | 57 | 0.43 (0.09–1.95) | 0.260 |
| | III | 123 | 0.45 (0.25–0.81) | 0.006 |
| | IV | 82 | 0.50 (0.27–0.94) | 0.029 |
| mTOR | I | 265 | 0.30 (0.12–0.77) | 0.008 |
| | II | 57 | 0.24 (0.06–0.86) | 0.018 |
| | III | 123 | 0.58 (0.33–1.02) | 0.053 |
| | IV | 82 | 1.36 (0.82–2.26) | 0.240 |

**Notes.**

[a] The total number was 527, because there were missing expression values and/ or incomplete survival data.

**Table 3** Correlation of the PI3K/AKT/mTOR mRNA expression with OS in different sex of ccRCC patients (K-M plotter, $n = 530$).

| Gene | Sex | n | HR (95% CI) | P-value |
|------|-----|---|-------------|---------|
| PTEN | female | 186 | 0.35 (0.21–0.59) | <0.001 |
| | male | 344 | 0.53 (0.36–0.77) | <0. 001 |
| PIK3CA | female | 186 | 0.27 (0.15–0.51) | <0. 001 |
| | male | 344 | 0.57 (0.39–0.82) | 0.003 |
| PIK3CB | female | 186 | 0.45 (0.27–0.76) | 0.002 |
| | male | 344 | 0.48 (0.33–0.70) | <0.001 |
| PIK3CD | female | 186 | 0.57 (0.34–0.97) | 0.037 |
| | male | 344 | 1.81 (1.22–2.68) | 0.003 |
| PIK3CG | female | 186 | 0.49 (0.30–0.82) | 0.006 |
| | male | 344 | 0.68 (0.47–0.99) | 0.044 |
| AKT1 | female | 186 | 0.51 (0.31–0.85) | 0.008 |
| | male | 344 | 0.76 (0.49–1.17) | 0.210 |
| AKT2 | female | 186 | 0.76 (0.46–1.25) | 0.270 |
| | male | 344 | 1.71 (1.15–2.55) | 0.007 |
| AKT3 | female | 186 | 0.32 (0.20–0.54) | <0.001 |
| | male | 344 | 0.48 (0.31–0.72) | <0.001 |
| mTOR | female | 186 | 0.27 (0.14–0.52) | <0.001 |
| | male | 344 | 0.78 (0.54–1.14) | 0.200 |

(*Han et al., 2017*). However, no significant correlation was observed between *PIK3CA, mTOR* and *p-mTOR (S2448)* expression and tumor stages or grades of patients with ccRCC ($P > 0.05$).

## DISCUSSION

ccRCC is the most prevalent kidney cancer; however, its molecular mechanism remains to be fully elucidated (*Han et al., 2017*). The PI3K/AKT/mTOR signaling pathway, which is a central regulator of cell survival and proliferation, is constitutively active in RCC and reportedly implicated in RCC pathogenesis and progression. In the present study, the mRNA expression levels of the aforementioned nine genes demonstrated no significant differences between ccRCC and normal tissues ($P > 0.05$), suggesting that the activation of the PI3K/AKT/mTOR signaling pathway in patients with ccRCC may not be dependent on their transcriptional regulation. Although WB displayed an upward trend of AKT protein expression in ccRCC, more samples were needed to clarify this issue. When striatified by clinical stage of the 523 ccRCC cases using GEPIA, the mRNA expression levels of six genes, i.e., *PTEN*, *PIK3CA*, *PIK3CB*, *AKT1*, *AKT2* and *AKT3*, were negatively correlated with their stages ($P < 0.05$). Further prognostic analysis using the K-M plotter and HPA demonstrated that the mRNA expression levels of the major PI3K/AKT/mTOR signaling pathway members (*PTEN, PIK3CA, PIK3CB, PIK3CG, AKT3* and *mTOR*) were positively correlated with a favorable OS in patients with ccRCC, whereas *PIK3CD* mRNA expression levels were positively correlated with a poor OS ($P < 0.05$). Accordingly, TCPA analysis demonstrated that the protein expression levels of PTEN, p-AKT (S473*)* and p-mTOR (S2448) were

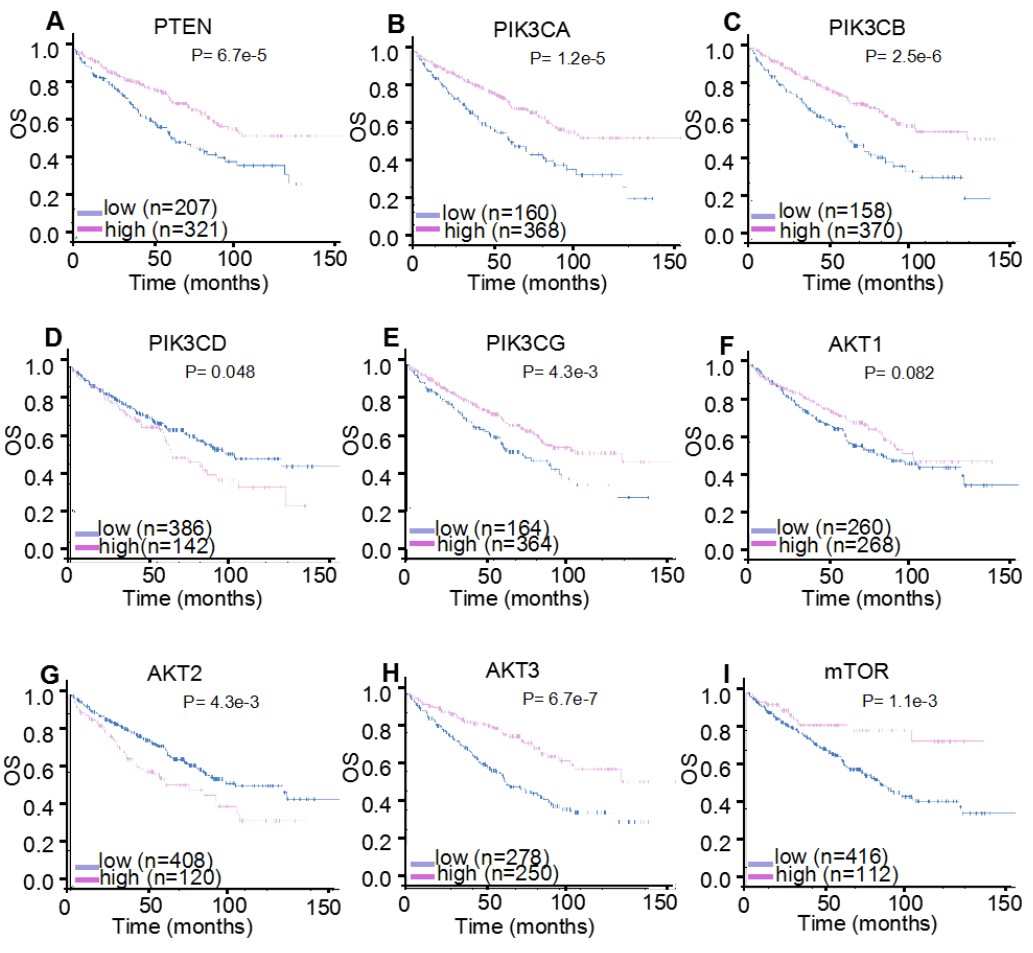

**Figure 6  HPA mRNA expression analysis displaying the overall survival curves of PI3K/AKT/mTOR signaling pathway members.** The overall survival curves of (A) *PTEN*, (B) *PIK3CA*, (C) *PIK3CB*, (D) *PIK3CD*, (E) *PIK3CG*, (F) *AKT1*, (G) *AKT2*, (H) *AKT3* and (I) *mTOR* in 528 patients with clear cell renal cell carcinoma.

correlated with an improved OS ($P < 0.05$). The relationship between their mRNA or protein expression levels and several clinicopathological features, such as sex, pathological stage and Fuhrman grade, were also analyzed. The results demonstrated that the majority of the stratified prognosis was consistent with the overall prognosis, which confirmed the validity of our conclusion. In general, the prognosis between the mRNA and protein expression levels of members of the PI3K/AKT/mTOR signaling pathway was consistent, with the majority being recognized as favorable prognostic factors in patients with ccRCC. The mRNA expression levels of the six genes, i.e., *PTEN*, *PIK3CA*, *PIK3CB*, *AKT1*, *AKT2* and *AKT3*, were negatively correlated with their stages, which was also consistent with their favorable prognosis in ccRCC. This suggested that the function of PI3K/AKT/mTOR in ccRCC was not only dependent on the mRNA and protein expression levels, but also reliant on protein activation through phosphorylation, such as *p-AKT* and *p-mTOR*. What's more, it also indicated that the activation of PI3K/AKT/mTOR was a complicated process, and

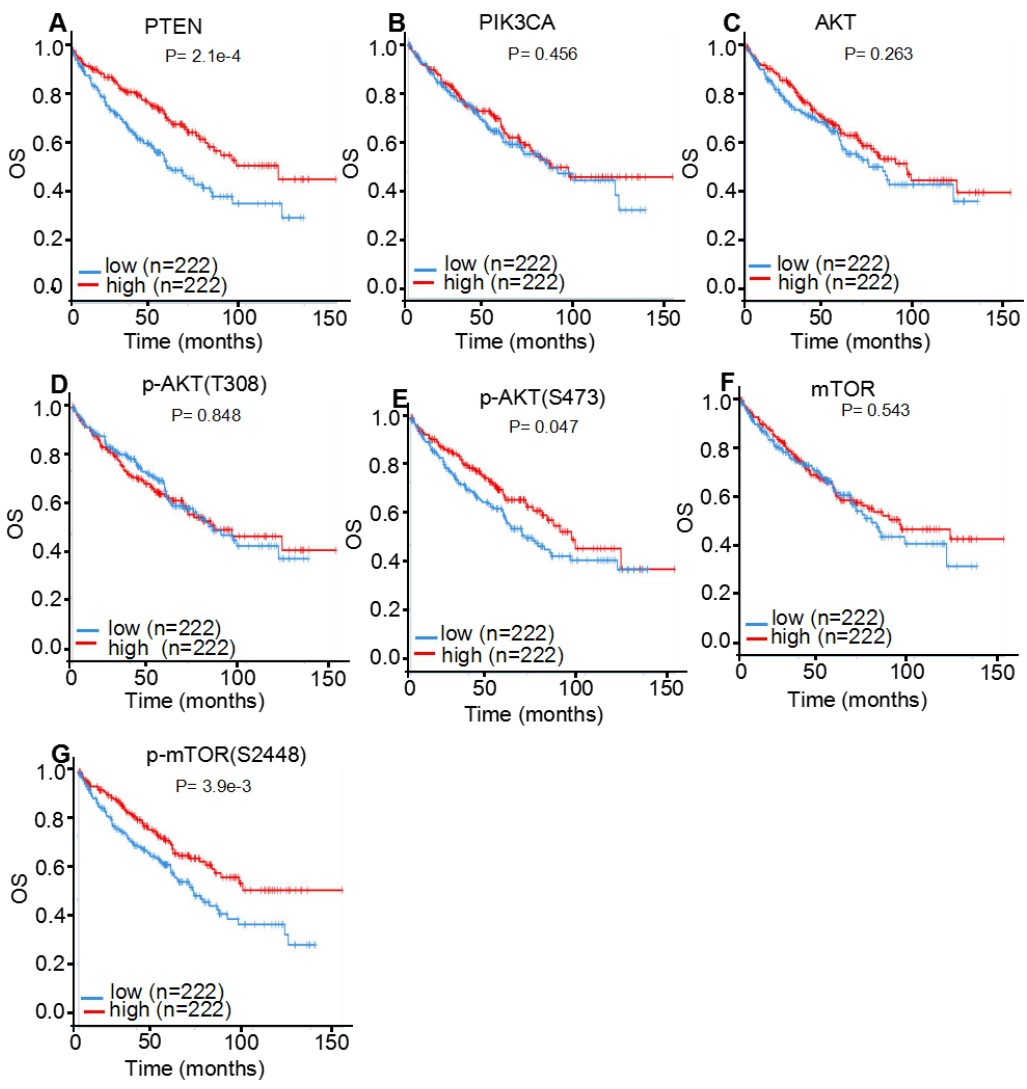

**Figure 7** **TCPA protein expression analysis displaying the overall survival curves of PI3K/AKT/mTOR signaling pathway members.** The overall survival curves of (A) *PTEN*, (B) *PIK3CA*, (C) *AKT*, (D) *p-AKT* (T308), (E) *p-AKT* (S473), (F) *mTOR* and (G) *p-mTOR* (S2448) in 445 patients with clear cell renal cell carcinoma.

further molecular biological, pathological and biochemical experiments were needed to support the prognosis of PI3K/AKT/mTOR pathway members in ccRCC. To the best of our knowledge, this was the first report studying the prognostic role of mRNA and protein expression levels of multiple PI3K/AKT/mTOR pathway members, such as *PTEN, AKT1* and *mTOR*, in ccRCC simultaneously. Our findings indicated that they may be potential prognostic markers for ccRCC, in addition to suggesting that PI3K/AKT/mTOR pathway signaling abnormalities may be involved in ccRCC tumorigenesis and progression.

The PI3K/AKT/mTOR signaling pathway serves an important role in ccRCC pathogenesis and progression, and is negatively regulated by the tumor suppressor *PTEN*.

**Table 4** Correlation of the PTEN/PI3K/AKT/mTOR protein expression with OS in ccRCC patients (TCPA, $n = 445$).

| Protein | Subgroup | Log-rank $P$ | Cox $P$[a] | Stage | Grade |
|---------|----------|------------|----------|-------|-------|
| PTEN | PTEN | <0.001 | <0.001 | <0.001 | <0.001 |
| PI3K | PIK3CA | 0.456 | 0.834 | 0.720 | 0.816 |
| AKT | AKT | 0.263 | 0.003 | 0.045 | <0.001 |
| | p-AKT(S473) | 0.047 | 0.002 | <0.001 | <0.001 |
| | p-AKT(T308) | 0.848 | 0.204 | 0.475 | 0.240 |
| mTOR | mTOR | 0.543 | 0.041 | 0.272 | 0.056 |
| | p-mTOR(S2448) | 0.004 | <0.001 | 0.062 | 0.147 |

**Notes.**

[a] Abbreviation for univariate Cox $P$.

PTEN has been identified as one of the most commonly lost or mutated tumor suppressor genes in human cancers (Que et al., 2018; Tang et al., 2017); however, although PTEN has been thoroughly investigated in RCC, its prognostic role remains controversial. For example, Zhu et al. (2015) reported that PTEN protein expression levels were decreased in RCC, and PTEN could be used as a favorable prognostic (OS) marker, whereas Kim et al. (2017) discovered that PTEN was not an independent prognostic marker of RCC. However, two meta-analyses indicated that PTEN was a favorable prognostic factor for patients with ccRCC (Que et al., 2018; Tang et al., 2017), which is consistent with our findings. In recent years, research has also focused on investigating the prognosis of other PI3K/AKT/mTOR signaling pathway proteins using immunohistochemistry analysis. For example, Merseburger et al. (2008) demonstrated that PI3K activation was inversely correlated with RCC patients' survival, and that low PTEN/high p-AKT expression levels were also associated with a decreased survival in 176 patients with RCC; Pantuck et al. (2007) found that high expression levels of nuclear p-AKT (S473) were associated with favorable disease-specific survival (DSS) in 375 patients with RCC; and Hager et al. (2009) reported that increased p-AKT (S473) expression levels were correlated with a poorer OS for 440 patients with RCC. In addition, Kruck et al. (2010) investigated the expression levels of mTOR and p-mTOR (S2448) in 10 ccRCC and normal kidney tissues, and found that p-mTOR (S2448), but not mTOR expression levels, were increased in ccRCC. A subsequent study by the same authors comprehensively investigated the differential expression levels of mTOR and p-mTOR in 342 primary and 90 metastatic ccRCC tissues, and revealed that high p-mTOR expression levels, but not mTOR levels, were associated with impaired OS (Rausch et al., 2019). Furthermore, Liontos et al. (2017) reported that post-treatment, the combination of increased p-mTOR expression levels and low VEGF expression levels was negatively correlated with OS in 79 patients with mRCC who were refractory to first-line sunitinib treatment. However, collectively these studies failed to reach a consistent conclusion and the prognostic role of the PI3K/AKT/mTOR signaling pathway in RCC remains controversial. The prognostic inconsistency of the PI3K/AKT/mTOR signaling pathway in RCC may be due to the remarkable heterogeneity of RCC specimens, the limited sample size and the divergence of clinicopathological backgrounds.

Integrated bioinformatics analysis confirmed that the PI3K/AKT/mTOR pathway was activated in ccRCC, but the prognostic significance was not fully elucidated (*Cancer Genome Atlas Research N, 2013*; *Chen et al., 2016*; *Han et al., 2017*; *Zhang et al., 2017*). Through pan-cancer proteogenomic atlas analysis, *Zhang et al. (2017)* found that PTEN, p-AKT and p-mTOR expression levels were all significantly correlated with improved outcomes in 32 major types of cancer, which was also observed in the present study. Similar results were reported by Han, who demonstrated that certain members of the PI3K/AKT/mTOR signaling pathway, such as PTEN protein expression, and PTEN and AKT3 mRNA expression were prognostic factors for ccRCC (*Han et al., 2017*). However, no clarification was provided on whether these were associated with a favorable or unfavorable prognosis (*Han et al., 2017*). In addition, in 446 ccRCC cases, integrative proteogenomic analysis revealed that some members of the PI3K/AKT/mTOR signaling pathway, for example, PTEN and AKT mRNA expression levels, and PTEN protein expression levels, were recognized as favorable predictors for ccRCC (*Cancer Genome Atlas Research N, 2013*), which was also consistent with our findings in the present study. Meanwhile, through multilevel genomics-based taxonomy of 894 RCC samples, *Chen et al. (2016)* demonstrated that the PI3K/AKT/mTOR signaling pathway could distinguish between RCC subtypes, such as clear cell, chromophobe and papillary RCC; however the prognostic value was not investigated. In the present study, comprehensive analysis of the prognostic significance of components of this pathway at the mRNA and protein level was performed in ~500 ccRCC tissues using the K-M plotter, HPA and TCPA databases; it was found that the majority of the mRNAs (such as *PTEN* and *AKT3*) and several proteins [such as *PTEN* and *p-AKT (S473)* ] were favorable prognostic factors for ccRCC, which indicated that the PI3K/AKT/mTOR signaling pathway may be implicated in ccRCC progression, which may also provide a potential therapeutic target for patients with ccRCC. These findings are contradictory to the common knowledge that the PI3K/AKT/mTOR signaling pathway promotes the initiation and progression of RCC. However, mTOR signaling is necessary for multiple biological effects, including energy metabolism homeostasis, autophagy, cell survival and apoptosis. Thus, the prognostic roles of PI3K/AKT/mTOR signaling pathway members in patients with RCC may be different under specific physiological or pathological conditions, although this hypothesis requires to be further investigated.

## CONCLUSIONS

The present study investigated the mRNA expression profiles of multiple PI3K/AKT/mTOR signaling pathway components, including *PTEN*, *PIK3CA*, *PIK3CB*, *PIK3CD*, *PIK3CG*, *AKT1*, *AKT2*, *AKT2* and *mTOR*, in ccRCC and normal tissues using GEPIA and Oncomine analyses, and detected PI3K, AKT and mTOR protein expression using WB analysis. The prognostic significance of these genes and proteins in ccRCC patients was evaluated using K-M plotter, HPA and TCPA analysis, which suggested that several members of the PI3K/AKT/mTOR signaling pathway, especially *PTEN*, may be favorable prognostic factors in ccRCC. To the best of our knowledge, our study was the first to reveal the detailed OS of nine PI3K/AKT/mTOR signaling pathway members simultaneously. These results further

suggested that the PI3K/AKT/mTOR signaling pathway may be implicated in ccRCC initiation and progression, and that components could be used as potential therapeutic targets in ccRCC.

### Funding
This work was supported by the Scientific Research Foundation of Shandong Provincial Qianfoshan Hospital (QYPY2019NFSC0601, Zuohui Zhao), the National Natural Science Foundation of China (81600124, Na Liu), and the Shandong Medical and Health Science and Technology Development Project (2016WS0481, Zuohui Zhao; 2016WS0484, Qiang Liu). The funders had no role in study design, data collection and analysis, decision to publish, or preparation of the manuscript.

### Grant Disclosures
The following grant information was disclosed by the authors:
Scientific Research Foundation of Shandong Provincial Qianfoshan Hospital: QYPY2019NFSC0601.
National Natural Science Foundation of China: 81600124.
Shandong Medical and Health Science and Technology Development Project: 2016WS0481, 2016WS0484.

### Competing Interests
The authors declare there are no competing interests.

### Author Contributions
- Demin Fan, Fei Wu, Na Liu and Hongyi Qu performed the experiments, authored or reviewed drafts of the paper, and approved the final draft.
- Qiang Liu performed the experiments, prepared figures and/or tables, and approved the final draft.
- Yijiao Yuan and Yong Li analyzed the data, authored or reviewed drafts of the paper, and approved the final draft.
- Huayu Gao, Juntao Ge, Yue Xu and Hao Wang analyzed the data, prepared figures and/or tables, and approved the final draft.
- Qingyong Liu analyzed the data, prepared figures and/or tables, authored or reviewed drafts of the paper, and approved the final draft.
- Zuohui Zhao conceived and designed the experiments, prepared figures and/or tables, authored or reviewed drafts of the paper, and approved the final draft.

### Human Ethics
The following information was supplied relating to ethical approvals (i.e., approving body and any reference numbers):

The First Affiliated Hospital of Shandong First Medical University granted Ethical approval to carry out the study (Ethical Application Ref: 2017-S007).

## Data Availability

The raw measurements are available in the Supplementary Files.

## Supplemental Information

Supplemental information for this article can be found online at http://dx.doi.org/10.7717/peerj.9261#supplemental-information.

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
