# Peer review of "Prognostic significance of PI3K/AKT/ mTOR signaling pathway members in clear cell renal cell carcinoma"

_PeerJ, doi:10.7717/peerj.9261_

## Round 0.1 · original submission · Major Revisions

Please respond to the reviewer comments.

thanks

dReviewer 1 ·

Basic reporting

no comment

Experimental design

K-M plotter and HPA prognostic analysis seems to do the same thing to the same TCGA dataset.

Validity of the findings

Please add the marker information, catalog number of the antibodies, secondary antibodies and the software/method to analysis the WB.

Additional comments

The manuscript entitled ‘Prognostic significance of PI3K/AKT/mTOR signaling pathway members in clear cell renal cell carcinoma’ describes the analysis of public TCGA ccRCC and GTEx normal tissue, including mRNA and protein expression to explore potential prognostic roles of PI3K/AKT/mTOR signaling pathway in ccRCC. They used GEPIA database to analyze mRNA expression level. Then they used a total of 3 databases to perform overall survival. The authors also experimentally demonstrated the expression of the highlighted potential markers using immunoblot. Though the conclusions seem to be valuable, I am not fully convinced by some details in the manuscript.

Major comment:
1. My major concern about GEPIA analysis is the batch effect, even though the correction is considered in GEPIA server. The GEPIA analysis in this study combined normal samples (n = 100) from both TCGA and GTEx. There are 72 matched normal samples in TCGA-KIRC. TCGA tumor versus TCGA normal only analysis should also be performed to validate the results. TCGA matched tumor versus matched normal may also be more convincing.
2. As mentioned in the “Experimental design”, both K-M plotter and HPA databases analysis generated the K-M plots. It seems that the authors replicated the analysis of the same dataset using two different databases. The difference would be number of patient samples involved in the analysis (n = 530 versus n = 528). Would you explain why both analyses are necessary?
3. Take HPA analysis as an example, what would be the cut-off for survival analysis? Please provide the analysis details.
4. In Figure 3-5, for each gene, patients were divided into two groups: high expression and low expression. Please list the exact patient number of high versus low for each gene.

Minor comment:
1. In line 119, ‘P-value were displayed’, however, only ‘P>0.05 Figure 1’ was mentioned without the exact numbers of each group.
2. In line 120, it mentioned that the unit of RNA-seq is TPM, which is correct. However, the unit in the figure is log2 normalized TPM. The unit should also be mention in the legend of Figure 1.
3. In Figure 1 legend, you may need to mention that the figures were from GEPIA server.
4. Please include molecular weight in Figure 2A (as shown in supplement materials).
5. Clinicopathological parameters was mentioned and listed for public dataset. Would you provide the information of the three samples used in immunoblotting?

Reviewer 2 ·

Basic reporting

no comment

Experimental design

1. mRNA level of PTEN, PIK3CA, PIK3CB, PIK3CG, AKT3 and mTOR should be tested by qRT-PCR to validate the bioinformatic analysis;
2. Western blot should be optimized.

Validity of the findings

no comment

Additional comments

This manuscript by Fan et al. reported the analysis of the mRNA expression levels of multiple PI3K/AKT/mTOR signaling pathway components, including PTEN, PIK3CA, PIK3CB, PIK3CD, PIK3CG, AKT1, AKT2, AKT2 and mTOR, in ccRCC and normal tissues based on GEPIA database, and detected PI3K, AKT and mTOR protein expression by WB. The analysis of these nine members at mRNA levels and proteins levels in ccRCC patients was investigated based on K-M plotter, HPA and TCPA. In general, the manuscript is clearly presented and written, and contains appropriate introductory material, methods, bioinformatic analysis and reasonable. As noted above, the entire study sounds entirely plausible and provides useful information for further research in the future. However, there are several major concerns as described as follows. I would suggest the paper being accepted for publication if the authors can incorporate these comments in their revision.
There are several concerns proposed as follows:

Major concerns:
1. There are a lot of analysis based on different databases with different outputs. For example, the GEPIA revealed no significantly different between ccRCC and normal tissues at the mRNA expression level. However, the analysis based on K-M plotter and HPA shows that PTEN, PIK3CA, PIK3CB, PIK3CG, AKT3 and mTOR were positively correlated with overall survival (OS), whereas PIK3CD mRNA expression was negatively correlated with OS. Please provided reasonable explanations of the completely different results. And I strongly suggest that you should check the mRNA level of PTEN, PIK3CA, PIK3CB, PIK3CG, AKT3 and mTOR by qRT-PCR to validate your analysis based on different database. Especially, PTEN should be tested.
2. Validation of P13K, AKT and mTOR protein by WB. There are nine members were investigated in this study, why do you just check these three members? And in Figure 2, the N2 has problems compared with N1 and N3, the Actin level of N2 is lower than Actin level of N1 and N3, but the mTOR lever is much higher in N2. For the Akt, T1 and T3 shows larger bands (maybe phosphorylated), but T2 doesn’t have this feature. Could you explain it? For the Supplementary file 1, the label on each lane is confusing, you’d better load sample orderly. If you use two blots, each blot should use Actin as a control. The marker is missing in mTOR blot. It will be better you labeled the Molecular weight of each bands of the marker. For the control, the amount of Actin should be the quite similar in each sample. Antibody information should be labeled for each protein.
Minor concerns:
1. In line 96, “(Han et al. 2017) discovered that numerous protein biomarkers” should be corrected as “Han et al (Han et al. 2017)discovered that numerous protein biomarkers”, the similar problems also happen in line 317, line 323 and line 334.
2. In Table 2, the font of I, II, and III should change into Times New Romans.
3. The Written Informed Consents in supplementary file 2 should be signed.

·

Basic reporting

no comment

Experimental design

no comment

Validity of the findings

no comment

Additional comments

The study presents a survival analysis of nine genes in renal cell carcinoma patients. Although, some of the genes were found to be correlated with the overall survival, the authors didn’t perform any additional analysis to explain why and in which cases this correlation might be significant. In addition, gene expression and protein level of the same factor was not significantly different between the tumor and normal tissue, which was not explained as well.

1. Despite being correlated with overall survival, the expression of the mRNA and level of the proteins of the markers were not different between cases and control conditions! Can this be explained? For example, the markers can be investigated on a granular level. There might be a pattern that is hidden in the full dataset. Gene expression and/or protein level could be different when stratified by clinical or demographic characteristics of the cases.
2. The section on “GEPIA database analysis” refers to the number and sources of the datasets and to the platform in which the data were processed. However, there is no mention on the origin of the data, the type of analysis performed or the criteria for significance.
3. The description of the databases and the platforms used in the study is confusing (Materials & Methods). GEPIA and KM plotter are not exactly database, but rather analysis platforms that rely on data generated from other projects, mainly TCGA. Please, consider describing the datasets separately from the type of analysis that was performed in each, then link them in a workflow/pipeline section.
4. It is not clear from the description of the workflow which datasets were used for testing/validation. Some of the analysis platforms rely on the same datasets, while presented as independent.
5. A univariate and multivariate analysis were not described in the methods section.

---

## Round 0.2 · Minor Revisions

Please revise the manuscript according to the reviewers' remaining comments.

Thanks

Reviewer 1 ·

Basic reporting

no comment

Experimental design

no comment

Validity of the findings

no comment

Additional comments

The authors have addressed my concerns in the revised version.
A minor comment: It would be better if F value could also be kept in Figure 3, because it is ANOVA analysis.

Reviewer 2 ·

Basic reporting

No comment.

Experimental design

No comment.

Validity of the findings

No comment.

Additional comments

This edition is much better. The authors corrected all of the proposed questions carefully and reasonably. It is awesome and suitable to be published.

·

Basic reporting

no comment

Experimental design

no comment

Validity of the findings

no comment

Additional comments

The authors responded to some of the issues raised in the review. They revised parts of the text to detail their methodology and added two figures; one to outline the workflow of the study and another to show the expression of the nine genes in different stages of the disease. The revised manuscript fails to address two critical issues.
1. In the revised manuscript, the authors show a correlation between six of the nine genes of interest and the tumor stage. However, the small effect size, the lack of comparison to control patient samples, the protein level comparison and the absent difference in three of the nine genes make it difficult to accept the difference in expression as a possible explanation for the suggested correlation.
2. The added figure (Figure 1) outlines the workflow of the study. The figure, and the text, fail however to respond to a serious issue. Some of the tools/platforms on which the study were performed use similar sources of data while they were presented in the manuscript as independent, as in some serves as discovery and others as validation.

---

## Round 0.3 · accepted · Accept

Thank you for a well-performed revision of the manuscript.